# A Semantic Cognition Contribution to Mood and Anxiety Disorder Pathophysiology

**DOI:** 10.3390/healthcare11060821

**Published:** 2023-03-10

**Authors:** Iván González-García, Maya Visser

**Affiliations:** Neuropsychology and Functional Neuroimaging Research Group, Department of Basic and Clinical Psychology and Psychobiology, University Jaume I, 12071 Castelló de la Plana, Spain

**Keywords:** mental health disorders, anxiety, depression, semantics, social, fMRI, anterior temporal lobes

## Abstract

Over the last two decades, the functional role of the bilateral anterior temporal lobes (bATLs) has been receiving more attention. They have been associated with semantics and social concept processing, and are regarded as a core region for depression. In the past, the role of the ATL has often been overlooked in semantic models based on functional magnetic resonance imaging (fMRI) due to geometric distortions in the BOLD signal. However, previous work has unequivocally associated the bATLs with these higher-order cognitive functions following advances in neuroimaging techniques to overcome the geometric distortions. At the same time, the importance of the neural basis of conceptual knowledge in understanding mood disorders became apparent. Theoretical models of the neural basis of mood and anxiety disorders have been classically studied from the emotion perspective, without concentrating on conceptual processing. However, recent work suggests that the ATL, a brain region underlying conceptual knowledge, plays an essential role in mood and anxiety disorders. Patients with anxiety and depression often cope with self-blaming biases and guilt. The theory is that in order to experience guilt, the brain needs to access the related conceptual information via the ATL. This narrative review describes how aberrant interactions of the ATL with the fronto–limbic emotional system could underlie mood and anxiety disorders.

## 1. Introduction

Depression and anxiety have been classically studied from the emotion perspective by concentrating on the disruption of emotional regulation in patients. However, recent research suggests that the neuroscience of conceptual processing is important in understanding these mental health disorders [1,2]. This review highlights a new role for the neural basis of semantic cognition in developing and treating mood and anxiety disorders. A recent review suggested that the right superior anterior temporal lobe (ATL) underlies the conceptual processing of social feelings, such as compassion and guilt [1]. Especially negative social feelings, such as guilt, play a role in mood and anxiety disorders [3,4,5], and understanding the functional role of the ATL in these processes will further clinical research and treatment development on this topic. As such, this narrative review concentrates on the role of the ATL in semantics, social feelings, and mental health disorders. The first part gives a general overview of the neural basis of mood and anxiety disorders. Subsequently, the neuroimaging literature that links the ATL to semantics and social conceptual processing, and as such plays an important role in experiencing social feelings such as guilt, is discussed. Next, we describe how aberrant interactions of the ATL with the fronto-limbic emotional system could underlie mood and anxiety disorders in patients that cope with self-blame and guilt feelings. Finally, this review gives future directions for investigating the possible role of the ATL in mood and anxiety disorders and other mental health disorders. 

### 1.1. The Neural Basis of Mood and Anxiety Disorders

In recent decades, neuroscience on mood and anxiety disorders has tried to identify biomarkers for diagnosis, prediction of treatment outcome, and treatment development [6,7]. Research has focused on the neural correlates of each of these mental pathologies with their symptomatology and some proposed etiologies such as emotional (dys)regulation. For example, an fMRI meta-analysis highlighted both distinct and overlapping functional and structural alterations for depression and anxiety disorders, supporting a possible transdiagnostic view [8]. Distinct regions for depression included the cingulum, medial frontal gyrus, precentral gyrus, amygdala, or hippocampus, whereas regions associated with anxiety included the orbitofrontal cortex, fusiform gyrus, anterior cingulate cortex, or insula. Overlapping regions for both disorders included certain areas of the anterior cingulate cortex, amygdala, hippocampus, or orbitofrontal cortex. These regions are associated with emotion processing and emotional regulation [8]. Aligning research comes from EEG data. Beck depression inventory (BDI) scores correlated with higher delta-band frequency connectivity in areas related to emotional processing (i.e., ventromedial amygdala, hippocampus, and right insula) [9]. Moreover, certain biomarkers in the alpha band are related to both anxiety and depression [10,11,12], this signature being more evident when considering the comorbidity of these disorders [13], reflecting the close relationship of mood and anxiety disorders. Aligning with this, emotional dysregulation is considered a causal factor for both mood and anxiety disorders, which is sustained by a shared cognitive profile for depression and anxiety patients during emotion-related experiences and regulatory strategies such as reappraisal or distancing strategies [14]. These factors are associated with altered functioning of amygdala–prefrontal cortical circuitries, although with mixed results regarding the exact pattern of interaction (i.e., coupling versus decoupling [15]). In addition, patients with anxiety and depression often cope with self-blaming biases and guilt. The theory is that in order to experience guilt, the brain needs to access the related conceptual information via the ATL. This narrative review describes how aberrant interactions of the ATL with the fronto-limbic emotional system could underlie mood and anxiety disorders with guilt and self-blame symptomology.

### 1.2. The Anterior Temporal Lobe as a Semantic Hub Region

To understand how the ATL, as a region underlying conceptual processing, plays a role in anxiety and mood disorders, it is important to first understand the background literature on the functional role of the ATL in concept processing. This topic has received more attention over the last few decades. A relatively new model is called the hub-and-spoke theory, which assigns an important function to the bilateral ATL in semantic processing (for review, see Lambon Ralph et al. [16]). The bilateral ATL is thought to underlie a transmodal hub region that combines modality-specific concept information distributed throughout the brain to form a unified transmodal semantic concept. This region is thought to be at the center of all daily tasks that require semantics, enabling the understanding of verbal and nonverbal conceptual information so that one can act accordingly. For example, to drink a cup of tea, one must recognize the related objects (cup, tea, and sugar) and interact properly (reaching and drinking). The bATL collaborates with modality-specific regions to do this. In more detail, bATL interactions with visual and praxis-related regions are needed for visual recognition and acknowledgement that reaching behaviors are required. This theory aligns with the pattern of damage seen in semantic dementia patients. These patients have bATL damage, which causes impairment of semantic tasks while other cognitive functions are preserved (episodic memory, sensory, and grammar skills), suggesting a unique role for this region in semantic processing [17,18,19,20]. 

However, until a decade ago, very little was known about the exact role of this region. This is due to the fact that most language models are often based on stroke patient and functional magnetic resonance imaging (fMRI) data. In contrast, ATL damage due to middle cerebral artery stroke is very rare [21]. Similarly, a meta-analysis showed that older neuroimaging studies (PET and fMRI) did not obtain a signal in the ATL, either due to distortion of the fMRI signal in this region and/or insufficient brain coverage, thereby omitting a large part of the ATL [22]. These methodological issues have led to the omission of the ATL in older language models based on these data types. However, the clear pattern of semantic impairments seen in semantic dementia patients [23] has led to the initial hypothesis that the ATL plays a key role in semantic processing. Indeed, over a decade of structural research using interdisciplinary methods including behavioral and distortion-corrected fMRI studies on healthy and clinical populations, TMS, and computational modelling has revealed the bATL as a transmodal hub for semantic processing [23,24,25,26,27,28,29,30,31,32,33], which was extensively described in a review article by Lambon Ralph et al. [16]. Furthermore, the use of newer imaging techniques and better MRI scanners ensures that the bATL is more commonly included in the data acquisition, e.g., [34,35]. 

These newer techniques have allowed research on the exact role of the left and right ATL in general and in social concept processing as well as exploring the subdivisions within these regions. According to the hub-and-spoke theory, a prominent feature of the bATL as a hub for semantic processing is the strong interaction between the left and right ATL. In contrast to theories assigning an absolute left–right distinction for processing verbal and nonverbal information, this model supports the idea of a functional gradient in which both ATLs are relevant for the processing of verbal and nonverbal information, albeit with a relative left-right difference, respectively [16,36,37]. Thus, the semantic hub comprises a heavily interconnected left and right ATL with overlapping yet distinctly different functions. This idea aligns with the findings of a large-scale meta-analysis [38] associating the bilateral ATL with semantic processing, with a left ATL predilection for verbal stimuli. In addition, computational modeling research suggests that this left–right-graded bilateral hub system reflects the semantic symptoms shown in SD patients with asymmetrical ATL damage [39]. Reflections of this functional gradient and the flexibility of this system are seen in the fact that damage to either ATL may be damped by an upregulation of the semantic function in the contralateral ATL [26,40,41]. However, mimicking this with transcranial magnetic stimulations (rTMS) gave mixed results, being successful in one study [42] but not in another [43]. 

Furthermore, previous work suggests that the ATL contains various subregions with different functions and functional connectivity patterns [24,32,44,45,46]. However, there is still a lack of consensus on the exact role of each of these subdivisions. Social conceptual processing has been associated with the right [47,48] and left [49,50] superior ATL. In agreement, results from an rTMS study showed that both left and right ATL might underlie social concept processing [51]. The temporal polar region has been associated with the conceptual processing of emotional stimuli [50]. However, the ventral ATL also seems to play a vital role in socioemotional concept processing [49,52], indicating that further research is required to unravel the exact role of these subdivisions. Furthermore, one might also need to take into account the alternative view that both semantic and social information are processed by the same structures, but social information leads to higher activations as it is more important for humans and essential for survival [53]. In other words, the authors suggest that socioemotional information is prioritized in the ATL without the need for a separate structure. This in contrast to the hub-and-spoke theory, which proposes that the inferior bATL is involved in all conceptual processing, whereas the superior bATL is more specialized in verbal and social information, albeit with a gradual, not absolute, difference [16,27,32,47]. 

From this literature, it is clear that the role of the ATL in social concept processing has received more attention over the last two decades. Whereas models on semantic processing tend to focus on general conceptual knowledge (i.e., recognizing a cat and knowing that it is a pet that meows), models on social concept processing focus on the neural basis of conceptual knowledge of social behavior. Further evidence that the ATL underlies social conceptual processing comes from patient data. ATL damage causes loss of person knowledge [54,55]. Moreover, patients with semantic dementia with cell loss in the right superior ATL exhibit disproportionate impairments in understanding social concepts compared with other abstract, but nonsocial, concepts [56], indicating that the superior part of the ATL plays a specific role in social conceptual processing. Based on these results, Olson et al. [55] suggest that although the superior ATL might not be strictly modular, it clearly is more sensitive to social concepts compared with other ATL regions. This social conceptual processing is essential when experiencing social feelings such as guilt, as this requires the interpretation of negative social behaviors or interactions. This attributes an important role to the ATL during these social feelings [1]. Overall, its function in social conceptual processing in addition to its connectivity to other limbic structures suggests that the ATL has an important role in modulating higher-level social behaviors [55]. 

### 1.3. Distorted Anterior Temporal Lobe Interactions Underlie Depression

It is exactly the interaction between the ATL and the limbic system that is of interest to researchers studying psychological disorders such as depression [5,48,57]. As explained in the first section, theoretical models of depression suggest that an important causal factor is the disruption of emotional regulation [58,59]. Patients have a reduced capacity to downregulate negative affect and are unable to regulate positive affect [60]. These factors are associated with altered functioning of amygdala-prefrontal cortical circuitries, although with mixed results on the exact pattern of interaction (i.e., coupling versus decoupling [15]). However, in addition to this amygdala–prefrontal circuitry, it has been suggested that neural regions involved in social concept processing play a vital role in depression [48,56]. In their work, they assumed that the understanding of the neuroscience of conceptual knowledge could be essential in understanding mental health disorders such as depression. They found that the superior ATL underlies social concept processing [47,56] and that this region is involved in the processing of values and moral emotions such as guilt [48]. Self-blaming biases and guilt are causally linked to depression [2,3,4,5,61], making it interesting to investigate the role of the ATL within the known amygdala–prefrontal circuitry involved in mood disorders. Indeed, this research group showed that excessive guilt, self-blame, and self-worthlessness in depressive patients are reflected in aberrant interactions between the ATL and the subgenual cingulate cortex (SCC) [2,5]. The authors suggest that in order to experience these social feelings, the brain needs to be able to process the conceptual knowledge of guilt and shame. Moreover, their theory is that it is the aberrant interaction between the social conceptual and emotional networks that is the key to understanding the neural basis of mood disorders (at least with respect to the guilt-based symptomology). However, the role of the ATL in the processing of social information and as a key region for mood disorders is mostly overlooked compared with the more robust research outcomes concentrating on the amygdala–prefrontal circuitry. As explained above, the omission of ATL regions in theoretical models of mood disorders is largely explained by the incapacity of the older standard fMRI protocols to obtain signals in this area. However, establishing the validity of ATL connectivity as a biomarker of symptoms is a valid research objective that can now be undertaken with the advances in neuroimaging techniques over the years. 

### 1.4. Distorted Anterior Temporal Lobe Interactions Underlie Anxiety

The ATL–limbic interaction is possibly also important for understanding anxiety disorders. Patients with anxiety disorders also cope with a disruption of emotional regulation. The reduced capacity to downregulate negative affect underlies both anxiety and depression, whereas the reduced ability to regulate positive affect may be more specific to depressive disorders [60]. As explained above, this is associated with the altered functioning of amygdala–prefrontal cortical circuitries. In more detail, trait anxiety increases the amygdala response and amygdala–prefrontal functional connectivity during the viewing of negative stimuli and/or emotional faces [62,63,64,65]. In addition, diffusion tensor imaging revealed that trait anxiety predicts weaker connections between the amygdala and the medial frontal regions [66]. A recent fMRI study showed that anxiety increased dynamic causal influences from the amygdala to the frontal lobe, which, in turn, was associated with decreased reappraisal abilities toward aversive stimuli [67]. In other words, anxiety affects amygdala–frontal interaction, thereby disturbing emotional regulation. However, it is not clear how ATL interaction with the amygdala and frontal regions is affected by (sub)clinical anxiety. 

As explained before, the excessive guilt, self-blame, and self-worthlessness in depressive patients are reflected in aberrant coupling between the ATL and the subgenual cingulate cortex (SCC) [2,5]. As anxiety patients cope with these same negative social feelings, it is likely that this also is underlined by aberrant bATL–limbic interactions. It is probably important to look beyond the ATL and the subgenual cingulate cortex interaction. Until now, it has not been clear which regions interact with the bATL during social–emotional processing and how these interactions underlie mood and anxiety disorders. One can imagine that the complex patterns seen in various mood and anxiety disorders will underlie a broader network of regions. Indeed, a few studies have revealed the functional connectivity (FC) of the ATL with limbic regions associated with emotional processes and/or (sub)clinical mental health problems. For example, Pantazatos et al. [68] found that decreased ATL–hippocampal FC was associated with social anxiety. Furthermore, while investigating the neural substrates of ruminative thoughts, which are an important factor in anxiety severity, Satyshur et al. [69] found that decreased left amygdala and right temporal pole (TP) FC was associated with increased rumination. However, contradictive findings make it difficult to understand the neural processing underlying these connections. Some studies found hyperconnectivity instead of hypoconnectivity of the ATL with limbic regions, which are findings that are difficult to align. For example, Li et al. [70] found increased TP–amygdala FC in participants with generalized anxiety disorder. Similarly, although Green et al. [5] found a decoupling between the ATL and the subgenual cingulate cortex (SCC) during guilt processing in depressive patients, a recent study found that it was actually increased ATL–SCC coupling that could be associated with the risk of reoccurrence of depressive episodes [2]. The authors suggested that, in general, aberrant interactions between these regions underlie depression, whether it is coupling or decoupling [2]. Overall, this indicates that although promising advances are being made in this field, further investigation is required to unravel the neural network of social concept processing and mental health problems and the role of the ATL. 

### 1.5. Using Neuromodulation as Treatment for Depression and Anxiety

One very promising aspect of this research field is the development of novel treatment options based on the theoretical knowledge of the differential coupling between these regions. Zahn et al. [6] developed a freely available software called Friend (for Functional Real-time Interactive Endogenous Neuromodulation and Decoding), which is based on real-time neurofeedback training. While in the scanner, the ATL–SCC coupling is displayed in real time during the retrieval of emotional autobiographical memories. This feedback is displayed as a visual thermometer in which increased temperature reflects increased coupling. Depressive patients were able to use this visual feedback to improve ATL–SCC coupling. Although further research is required to investigate the impact of this neural training on clinical depression measures, it is a very promising clinical tool. In addition, the use of this technique could be generalized to various clinical profiles such as anxiety and other mental health disorders. The neural coupling of the ATL with the limbic regions is an important aspect of treatment, which is clear from a study on social anxiety [68]. Pantazatos et al. [68] found that reduced ATL–hippocampal FC during face processing could discriminate individuals with social anxiety from healthy controls. Moreover, after eight-week medical treatment with paroxetine, ATL–hippocampal FC increased. This shows the importance of understanding the functional interaction of the ATL with the limbic structures and related regions. Overall, these studies suggest that understanding the neural substrates involved in personal well-being has important implications for treatment development. 

### 1.6. Patients with Mood and Anxiety Disorders without Symptoms of Self-Blame and Guilt

It is important to note that the current review focusses on the neural basis of guilt feelings in patients with anxiety and depressive disorders. As such, it does not focus on patients as a homogenous group, but rather on specific symptomology that is present in most, but not all, patients. However, as a previous review suggested, research on this topic might not be possible if the aim is to study patients as homogenous groups [71]. Their review on clinical neuroscience depicted several important factors that should be taken into account when studying mental health disorders such as anxiety and depression. According to Hitcock et al. [71], one of the problems is that some studies treat mental health diseases as homogeneous, comparing patients with controls with the aim of finding clear biomarkers that can then be addressed with drug therapy. Although some diseases might have clear biomarkers (e.g., Tic disorders), mental health diseases such as anxiety and depression arise from interacting factors including genetics, brain changes, environment, and time [71]. Studying these mental health disorders as homogeneous groups therefore seems illogical. A solution would be to concentrate on individualized therapy based on the symptoms of each patient. Cognitive neuroscience research on symptomology would therefore be a logical step to aid treatment development. In other words, instead of comparing depressive patients with controls, research should focus on smaller subgroups, such as patients with increased levels of guilt and self-blame. Studying this specific symptomology could, in return, inform clinicians focusing on the development of individualized therapy. Of course, it will need to be clarified that the resulting outcomes and treatment developments are only useful in a subset of these patient groups (i.e., patients with feelings of self-blame and guilt). 

## 2. Factors to Take into Account for Research on the Role of the ATL in Mental Health Disorders

The economic costs of brain disorders (including mental disorders) in Europe are high [72]. In addition to direct health care costs, indirect costs (e.g., absenteeism from work, pensions, etc.) form an important problem for patients [72]. Furthermore, both mood and anxiety disorders are highly prevalent in the population, especially after the COVID-19 pandemic [73,74], forming a substantial economic burden to society [75]. Therefore, research on mood and anxiety disorders is vital for new insights about the diseases as well as rehabilitation. The current review suggests including the ATL as an important focus for future research on this topic. The final part of this review emphasizes some factors that need to be taken into account when studying mental health disorders. 

### 2.1. Sex Differences

Research has shown significant sex differences in the prevalence of mood and anxiety disorders, with predominant prevalence among women in adult age groups [76,77]. Possible factors contributing to these sex differences are genetics, hormones, and heightened exposure to severe adversity (particularly sexual abuse and violence) [76,78]. However, despite these clear sex differences, this factor is often excluded in clinical, pharmaceutical, and neuroscience research and, consequently, in treatment development [79]. With respect to the role of the ATL in guilt feelings, sex differences are an important factor as women tend to report stronger feelings of guilt [80]. As such, new projects on this topic should include gender as a variable, thereby ensuring that the resulting theoretical model is able to shed light on these differences and that future treatment developments will take this into consideration. Nowadays, large databases including DTI and resting-state images exist that allow the exploration of sex differences in functional and structural connectivity within the ATL–limbic–frontal neural network and its interaction with anxiety and depression measures (e.g., http://www.humanconnectomeproject.org/; accessed on 2 February 2023). Understanding gender differences is important from a treatment perspective, especially with new upcoming techniques such as real-time neurofeedback training. These enable individuals to improve the coupling between two regions that are identified as key regions implicated in mood disorders with the goal of decreasing symptoms [6]. If critical gender differences are revealed, they should be taken into account for such specific treatment options.

### 2.2. Variety of Clinical and Subclinical Mental Health Disorders

As explained above, most of the work investigating the role of the ATL on mental health disorders has focused on depression, specifically major depressive disorder [1,6,81]. In addition, this short review focusses on its possible role in anxiety disorders. However, there are many different types of mood and anxiety disorders in addition to the subclinical levels of these diseases [82]. Mood and anxiety disorders are spectral in nature; it is therefore also important to examine symptomatology in individuals without a clinical diagnosis to create a full understanding of these disorders [62,82]. Moreover, as individuals with subclinical levels may share common symptomatology, more fully understanding the subclinical populations will also lead to a better understanding of diagnosed individuals. 

Moreover, it is plausible that the bATLs have an important role in many mental health disorders that deal with negative or inappropriate social feelings. Abnormal social relationships are a common factor in most mental health issues [83]. For example, people with autism have impaired social behavior [84]. Similarly, patients with obsessive-compulsive disorder (OCD) are affected by social comparisons, indicating abnormal functioning in social contexts [85]. Taking the narrative of this short review into account, that would suggest an important role for the bATLs in these disorders. Therefore, future research could focus on the role of the ATL in negative social feelings in a variety of mental health disorders at clinical and subclinical levels. 

### 2.3. The Use of Advanced fMRI Techniques

As explained throughout this review, older models of semantic processing based on fMRI overlooked the ATL either due to distortion or loss of the BOLD signal or due to insufficient brain coverage [22]. However, advances in fMRI techniques over the last decade have allowed researchers to obtain signal from this region by using multiband and multi-echo imaging [34,35]. Indeed, this has led to a renewed interest in the role of the ATL in semantics, social behaviors, and mental health disorders [16,81,86]. Therefore, the use of these advanced imaging techniques leads the way to novel research on these topics. 

## 3. Final Remarks and Conclusions

Understanding the neural signature of mental health problems such as anxiety and depression will open new pathways to clinical treatment and prevention. As explained above, mental health disorders are highly prevalent in the population, creating a global public health problem. In addition to direct health care costs, indirect costs (e.g., absenteeism from work, pensions, etc.) present an important problem for patients [72]. Therefore, research on mental health issues is vital for new insights about these diseases as well as rehabilitation. The role of the ATL has long been overlooked as a key region in mood and anxiety disorders. However, advances in imaging techniques enable mapping of the functional roles of the ATL subregions in these disorders. This will have important implications for the creation of theoretical clinical models of mental health disorders in addition to influencing treatment development. For example, neural regions that show a clear pattern of decoupling associated with a mental health disorder can be targeted for real-time neural coupling training as a treatment option [6]. Therefore, understanding the functioning of bATL–fronto–limbic interactions will have important implications for clinical research. More specifically, establishing the validity of ATL connectivity as a biomarker of symptoms and treatment efficacy should be an important motivator for future research. 

## Data Availability

Data sharing not applicable.

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
