# Peer review of "A Semantic Cognition Contribution to Mood and Anxiety Disorder Pathophysiology"

_healthcare, 2023, doi:10.3390/healthcare11060821_

Round 1
Reviewer 1 Report
This paper seeks to explain the role the ATL play in mood and anxiety disorders, and how semantic cognition is involved in developing and treating mood disorders. This is a fine goal given the recent advancements in technology and ability to further understand the ATL, but there are a few major issues that will need to be addressed.
Major comments:
1. One of the conceptual aspects I could use more clarity on is the focus between bATL and unilateral ATL, as these are discussed throughout like they are being used interchangeably. For example, starting in the abstract the bATL is discussed, then it switches to the ATL without mention of right, left, or both lobes. It’s also discussed as this is only one region [bilateral anterior temporal lobe, not lobeS, as bilateral should imply). This needs to be further parsed out.
2. The ATL in general is a large region of the brain. What area or areas specifically are you discussing? For example, are these the temporal poles, or Brodmann’s area 38, superior/inferior/etc? I’m concerned that the roles you discuss are not associated with overlapping regions, and if they are not this should be discussed.
3. The research on mood and anxiety disorders extends far out from emotion regulation. There is a lot of research simply examining the correlates of each with both diagnosis and symptomology. I would suggest the authors take a deeper look into this literature to correct this background discussion in the intro.
4. Mood and anxiety disorders are discussed like they are related disorders. They are highly comorbid, but still two distinct disorders with differing brain activity patterns (see Knyazev and colleagues, 2016; Chandler et al., 2022; Manna et al., 2010; Feldmann et al., 2018; Adolph & Margraf, 2017 ).
5. The connection discussed between semantic dementia with the ATL does not connect how the ATL connects with mood disorders in the text.
6. In discussing mood disorders, only several behaviors are described in how they connect with the ATL - e.g. guilt and self-blaming. Guilt is only one of the symptoms (and not in all countries), and self-blaming is not particularly linked as a specific symptom of depression. The link therefore described between ATL and depression, or the wider concept of mood disorders, here is weak. The authors also don’t discuss how the ATL is related with people experience mood disorders without these specific symptoms.
7. The title and aim of the paper is related to the connection between the ATL with mood and anxiety disorders. The narrative however on the connection with anxiety disorders revolves around future directions, and open questions for which to pursue. I’m not sure this matches the title or proposed aim of the study discussed in the abstract, as it appeared the authors were going to give a review of the literature that supports this connection.
8. The introduction focuses on three aspects:
1. Semantics
2. Social interactions
3. Mental health disorders
Yet the role social interactions play in mood and anxiety disorders does not get mentioned again, at least not in a way that adds to the paper.
9. According to the Healthcare journal, this paper should have a minimum of 4000 words. A word count of the written product from the introduction to the end of the discussion is less than 3000 words. I’m not sure this meets minimum requirements for a review paper in this journal.
10. Overall the review itself is very much ‘on the surface’, without providing much more than a suggestion that bilateral ATLs play a role in anxiety and depression. There is no in-depth discussion about either type of disorder with the ATL, and no explanation why these two different disorders are included in the same paper.
Minor Comments
The writing in general is good, but there are a few typos. For example, there are typos of numbering throughout. e.g. - 1.1. is listed instead of 1.2, more than once.
Line 65 “an unique role” should be “a unique role”
The intro begins with the economic burden of mental disorders, which isn’t linked with the aim of the paper. Additionally, this paper doesn’t add how it can help address this issue. In other words, how does knowing about the ATL help with this burden?
The visuospatial integration (e.g. reaching for a cup in line 60) involves the posterior parietal communication with the premotor cortex as well. The mention of the ATL in reaching is lacking thorough explanation.
Line 155 states that “until now it is not clear …”, but then proceeds to state that future research is required. This is contradictory, which occurs a couple other times in the paper.
The paragraph about gender differences seems tangental when the rest of the paper is about semantics with mood and anxiety disorders.
Author Response
Comments from Reviewer 1
Comment 1: One of the conceptual aspects I could use more clarity on is the focus between bATL and unilateral ATL, as these are discussed throughout like they are being used interchangeably. For example, starting in the abstract the bATL is discussed, then it switches to the ATL without mention of right, left, or both lobes. It’s also discussed as this is only one region [bilateral anterior temporal lobe, not lobeS, as bilateral should imply). This needs to be further parsed out.
Response: We agree with this observation. As a consequence, we have added a paragraph explaining the relationship between the left and right ATL and their relevance for the bATL semantic hub (lines 114 - 130).
Comment 2: The ATL in general is a large region of the brain. What area or areas specifically are you discussing? For example, are these the temporal poles, or Brodmann’s area 38, superior/inferior/etc? I’m concerned that the roles you discuss are not associated with overlapping regions, and if they are not this should be discussed.
Response: We agree that it is convenient to fill this gap. We have included information relative to the literature regarding the function of the different subdivisions of the ATL (lines 132-145).
Comment 3: The research on mood and anxiety disorders extends far out from emotion regulation. There is a lot of research simply examining the correlates of each with both diagnosis and symptomatology. I would suggest the authors take a deeper look into this literature to correct this background discussion in the intro.
Response: We agree with this observation. We have added a paragraph explaining that the current review focusses on the specific symptoms of self-blame and guilt, but that this is just a small part of the symptomology of these patient groups (section 1.6, line 265-285).
Comment 4. Mood and anxiety disorders are discussed like they are related disorders. They are highly comorbid, but still two distinct disorders with differing brain activity patterns (see Knyazev and colleagues, 2016; Chandler et al., 2022; Manna et al., 2010; Feldmann et al., 2018; Adolph & Margraf, 2017 ).
Response: We agree on the fact that further emphasis on the neural correlates of the mentioned disorders is requires. We have included the recommended literature. In addition, we have added an fMRI metanalysis to cover the observed gap (section 1.1, lines 49 – 77).
Comment 5: The connection discussed between semantic dementia with the ATL does not connect how the ATL connects with mood disorders in the text.
Response: We agree that that might not have been clear in the first version of the manuscript. Our aim of this section of the manuscript is to cite the existing evidence relating the ATL to semantic processing whether referring to general semantics or social concepts. ATL damage in these patients is associated with pure semantic problems assigning an important role to the ATL during semantic processing. We have now introduced the objective of this section more clearly by adding: “To understand how the ATL, as a region underlying conceptual processing, plays a role in anxiety and mood disorders, it is important to first understand the background literature on the functional role of the ATL in concept processing. This topic has received more attention over the last few decades. “ at the beginning of the section (line 80-82).
Comment 6: In discussing mood disorders, only several behaviors are described in how they connect with the ATL - e.g. guilt and self-blaming. Guilt is only one of the symptoms (and not in all countries), and self-blaming is not particularly linked as a specific symptom of depression. The link therefore described between ATL and depression, or the wider concept of mood disorders, here is weak. The authors also don’t discuss how the ATL is related with people experience mood disorders without these specific symptoms.
Response: We agree with this observation. To answer this specific aspect we added the following paragraph (line 265 -286).
1.6. Patients with mood and anxiety disorders without symptoms of self-blame and guilt. It is important to note that the current review focusses on the neural basis of guilt feelings in patients with anxiety and depressive disorders. As such, it does not focus on patients as a homogenous group, but on specific symptomology that is present in most but not all patients. However, as a previous review suggests, research on this topic might not be possible if the aim is to study this as homogenous groups. Their review on clinical neuroscience depicted several important factors that should be taken into account when studying mental health disorders susch as anxiety and depression71. According to Hitcock et al. 71, one of the problems is that some studies treat mental health diseases as homogeneous, comparing patients with controls with the aim is to find clear biomarkers that can then be adressed with drug therapy. Although some diseases might have clear biomarkers (e.g., Tic disorders), mental health diseases such as anxiety and depression arise from interacting factors including genetics, brain changes, environment and time71. Studying these mental health disorders as homogeneous groups therefore seems illogical. A solution would be to concentrate on individualised therapy based on the symptoms of each patient. Cognitive neuroscience research on the symptomology would therefore be a logical step to aid treatment development. In other words, instead of comparing depressive patients with controls, research should focus on smaller sub-groups, suchs as patients with increased levels of guilt and self-blame. Studying this specific symptomology could in return inform clinicians focussing on the development of individualised therapy. Of course this means that it will need to be clarified that the resulting outcomes and treatment developments are only usefull in a subset of these patients groups (i.e., patients with feelings of self-blame and guilt).
Comment 7.The title and aim of the paper is related to the connection between the ATL with mood and anxiety disorders. The narrative however on the connection with anxiety disorders revolves around future directions, and open questions for which to pursue. I’m not sure this matches the title or proposed aim of the study discussed in the abstract, as it appeared the authors were going to give a review of the literature that supports this connection.
Response: We’ve addressed this issue by adding papers on this topic. First, we added studies that find aberrant functional connectivity in the limbic system associated with anxiety (line 2015-213). Second, we added papers that find aberrant ATL connectivity with the limbic system due to anxiety (Li et al., 2016; Pantazatos et al., 2014) or associated with anxiety symptomology such as rumination (Satyshur et al.2018) (line 214-236).
Comment 8: The introduction focuses on three aspects:
1. Semantics
2. Social interactions
3. Mental health disorders
Yet the role social interactions play in mood and anxiety disorders does not get mentioned again, at least not in a way that adds to the paper.
Response: We have changed the Introduction to focus on: 1) semantics, 2) social feelings and 3) mental health disorders. As such we have focused more specifically on social feelings such as guilt. Guilt requires the interpretation of negative social behaviours or interactions and Eslinger et al.(2019) suggest that this relies on the right superior ATL. Social feelings and in particular guilt, play an important role in anxiety and depression. As such, the ATL should play an important role in depression and anxiety in patients with self-blam and guilt feelings. (lines 39-45 and 161-166).
Comment 9: According to the Healthcare journal, this paper should have a minimum of 4000 words. A word count of the written product from the introduction to the end of the discussion is less than 3000 words. I’m not sure this meets minimum requirements for a review paper in this journal.
Response: We agree and have added sections. The word count is now over 4000.
Comment 10: Overall the review itself is very much ‘on the surface’, without providing much more than a suggestion that bilateral ATLs play a role in anxiety and depression. There is no in-depth discussion about either type of disorder with the ATL, and no explanation why these two different disorders are included in the same paper.
Response: We have now added a more in depth section on the role of the ATL in anxiety (see comment 7). In this section we also explain that both mental health disorders are associated with feelings of self-blame and guilt and it is this aspect that is the interest of the current paper.
Minor comments Reviewer 1
Minor comment 1: The Reviewer commented on some typos.
Response: We have corrected these errors.
Minor comment 2: The intro begins with the economic burden of mental disorders, which isn’t linked with the aim of the paper. Additionally, this paper doesn’t add how it can help address this issue. In other words, how does knowing about the ATL help with this burden?
Response: We agree with the Reviewers and deleted this section from the Introduction and added it at the end of the paper to suggest that future research on this topic is essential (lines 291-299).
Minor comment 3. The visuospatial integration (e.g. reaching for a cup in line 60) involves the posterior parietal communication with the premotor cortex as well. The mention of the ATL in reaching is lacking thorough explanation.
Response: We agree that this was unclear. Praxis related regions are not needed for reaching. However, if you want to drink tea, you first need to recognise the teacup and understand that it involves drinking. You need conceptual knowledge to understand that drinking tea requires reaching for the cup and bringing it to your mouth. We have rewritten this information to be more clear (line 92-93):
“In more detail, bATL interactions with visual and praxis-related regions are needed for visual recognition and acknowledgement that reaching behaviors are required”.
Minor comment 4. Line 155 states that “until now it is not clear …”, but then proceeds to state that future research is required. This is contradictory, which occurs a couple other times in the paper.
Response: We are not sure we understand the problem here. If something is unclear, then future research is required to clarify inconsistencies.
Minor comment 5. The paragraph about gender differences seems tangental when the rest of the paper is about semantics with mood and anxiety disorders.
Response: This section is meant to only address some factors that need to be taken into account for future research on this topic. It is therefore meant to be tangential. However, we hope that by restructuring the paper, this now has become clearer.
Reviewer 2 Report
I was asked to review a Review type manuscript, so expected a standard critical review piece with a fair comparison of multiple studies, preferably using a big table and/or a model. However, the author writes twice this is a "short narrative review", which is a format I am not familiar with. Therefore my views might be less useful in this case, as I am not familiar with the format and particularly could not understand the rationale.
It is a well-written manuscript summarizing familiar results from one lab (and it's alumni), results that have been published and summarized many times in the past. There is no effort to locate contrasting findings, as I would expect a review to do. For instance - Wong, C., & Gallate, J. (2011). Low-frequency repetitive transcranial magnetic stimulation of the anterior temporal lobes does not dissociate social versus nonsocial semantic knowledge. Quarterly Journal of Experimental Psychology, 64(5), 855-870; Bonnì, S., Koch, G., Miniussi, C., Bassi, M. S., Caltagirone, C., & Gainotti, G. (2015). Role of the anterior temporal lobes in semantic representations: paradoxical results of a cTBS study. Neuropsychologia, 76, 163-169.; Important previous reviews were not included, maybe because they do not fit the main hypothesis? for example, Wong, C., & Gallate, J. (2012). The function of the anterior temporal lobe: a review of the empirical evidence. Brain research, 1449, 94-116. The ideas regarding social roles and the ROI were already presented by Olson, I. R., McCoy, D., Klobusicky, E., & Ross, L. A. (2013). Social cognition and the anterior temporal lobes: a review and theoretical framework. Social cognitive and affective neuroscience, 8(2), 123-133. Perhaps the author can highlight the novelty of this manuscript? and try to evaluate contrasting possibilities. Having said the above, I should say I see strength points in the ms, and the final paragraphs are interesting and might inspire future research. However I think more effort should be invested in generating a more balanced reviewAuthor Response
Comments from Reviewer 2
Comment: I was asked to review a Review type manuscript, so expected a standard critical review piece with a fair comparison of multiple studies, preferably using a big table and/or a model. However, the author writes twice this is a "short narrative review", which is a format I am not familiar with. Therefore my views might be less useful in this case, as I am not familiar with the format and particularly could not understand the rationale.
It is a well-written manuscript summarizing familiar results from one lab (and it's alumni), results that have been published and summarized many times in the past. There is no effort to locate contrasting findings, as I would expect a review to do. For instance - Wong, C., & Gallate, J. (2011). Low-frequency repetitive transcranial magnetic stimulation of the anterior temporal lobes does not dissociate social versus nonsocial semantic knowledge. Quarterly Journal of Experimental Psychology, 64(5), 855-870; Bonnì, S., Koch, G., Miniussi, C., Bassi, M. S., Caltagirone, C., & Gainotti, G. (2015). Role of the anterior temporal lobes in semantic representations: paradoxical results of a cTBS study. Neuropsychologia, 76, 163-169.; Important previous reviews were not included, maybe because they do not fit the main hypothesis? for example, Wong, C., & Gallate, J. (2012). The function of the anterior temporal lobe: a review of the empirical evidence. Brain research, 1449, 94-116. The ideas regarding social roles and the ROI were already presented by Olson, I. R., McCoy, D., Klobusicky, E., & Ross, L. A. (2013). Social cognition and the anterior temporal lobes: a review and theoretical framework. Social cognitive and affective neuroscience, 8(2), 123-133. Perhaps the author can highlight the novelty of this manuscript? and try to evaluate contrasting possibilities. Having said the above, I should say I see strength points in the ms, and the final paragraphs are interesting and might inspire future research. However I think more effort should be invested in generating a more balanced review
Response: We thank the Reviewer for his/her opinion and the suggested papers have been included (lines 130-131, 136-137, and 142-147). In addition to the changes made as requested by Reviewer 1, we are confident that this results in a more balanced review.
Reviewer 3 Report
I congratulate the authors for the relevant topic. In the area of study there is little consensus for the findings summarized in the text, and there is a need for further studies.
Major comments:
In view of the many gaps for fronto-limbic system connections in mood and anxiety disorders, why not do a systematic review focused on a well-defined and delimited question, which may or may not have analysis of various outcomes relevant to the scope of the study ?
As this is a narrative review, the inferences assumed have little robustness, and validation is necessary (by more adjusted methods: such as well-defined eligibility criteria, peer analysis in relation to the selection, data extraction and analysis of the main findings in studies; risk of bias analysis; assessment of the level of evidence). In this context, I cannot accurately rate the study.
Unfortunately my opinion is not favorable to the publication.
Sincerely
Author Response
Comments from Reviewer 3
Comment: I congratulate the authors for the relevant topic. In the area of study there is little consensus for the findings summarized in the text, and there is a need for further studies.
Major comments:
In view of the many gaps for fronto-limbic system connections in mood and anxiety disorders, why not do a systematic review focused on a well-defined and delimited question, which may or may not have analysis of various outcomes relevant to the scope of the study ?
As this is a narrative review, the inferences assumed have little robustness, and validation is necessary (by more adjusted methods: such as well-defined eligibility criteria, peer analysis in relation to the selection, data extraction and analysis of the main findings in studies; risk of bias analysis; assessment of the level of evidence). In this context, I cannot accurately rate the study.
Response: Indeed this is a narrative review and as such we have not used a systematic literature search. We hope that the changes made as a response to Reviewer 1 and 2 are satisfactory to the Reviewer in that it provides a more balanced view on the current literature on this topic.
Round 2
Reviewer 3 Report
Comments to the Author
I read the present manuscript entitled “A semantic cognition contribution to mood and anxiety disorder pathophysiology” with a major interest. In this review the colleagues identified a gap in the literature. The authors combined a panel of investigations to decipher the functional impact of the bilateral Anterior Temporal Lobes (bATL) in the mood and anxiety disorders. The findings have been associated with semantics, social concept processing and even as a core region for depression.
The improvement in the presentation of the manuscript is evident. The manuscript is clearly presented and highlights all the novelty of this work. I am in favor of publication.
Best wishes!